# Review of Piezoelectric Micromachined Ultrasonic Transducers for Rangefinders

**DOI:** 10.3390/mi14020374

**Published:** 2023-02-02

**Authors:** Jiong Pan, Chenyu Bai, Qincheng Zheng, Huikai Xie

**Affiliations:** 1School of Integrated Circuits and Electronics, Beijing Institute of Technology (BIT), Beijing 100081, China; 2BIT Chongqing Center for Microelectronics and Microsystems, Chongqing 400030, China

**Keywords:** pMUT, piezoelectric, rangefinder, range-finding, microelectromechanical system, MEMS

## Abstract

Piezoelectric micromachined ultrasonic transducer (pMUT) rangefinders have been rapidly developed in the last decade. With high output pressure to enable long-range detection and low power consumption (16 μW for over 1 m range detection has been reported), pMUT rangefinders have drawn extensive attention to mobile range-finding. pMUT rangefinders with different strategies to enhance range-finding performance have been developed, including the utilization of pMUT arrays, advanced device structures, and novel piezoelectric materials, and the improvements of range-finding methods. This work briefly introduces the working principle of pMUT rangefinders and then provides an extensive overview of recent advancements that improve the performance of pMUT rangefinders, including advanced pMUT devices and range-finding methods used in pMUT rangefinder systems. Finally, several derivative systems of pMUT rangefinders enabling pMUT rangefinders for broader applications are presented.

## 1. Introduction

Piezoelectric micromachined ultrasonic transducers, or pMUTs, are microelectromechanical system (MEMS) devices that convert electrical energy to mechanical energy or vice versa through the flexural vibration (or deflection) of a membrane composed of a thin piezoelectric film based on piezoelectric effects [1,2,3]. A system with a single or several pMUTs can be used to transmit [4], to receive [5], or to both transmit and receive ultrasound signals (e.g., a pMUT imager [6,7], flowmeter [8], or rangefinder [9,10]), as illustrated in Figure 1. This article is focused on pMUT rangefinders. A pMUT rangefinder takes advantage of the signal transmitting and receiving capability of pMUTs to obtain a range between the target and the device [11], containing a single pMUT, multiple pMUTs, or an array of pMUTs to perform both ultrasonic transmitting and receiving. A pMUT rangefinder is only responsible for finding the range of a target but not the angle and direction of the target; derivative systems made of groups of pMUT rangefinders are able to perform more complicated tasks such as target positioning and object detection based on a group of range data [12,13]. It should be noted that pMUT rangefinder systems are different from pMUT imaging systems in that the objectives of pMUT imaging systems are to acquire the spatial distribution or three-dimensional (3D) structural information of a target but not the distances between the pMUT and the target points [14]. 

pMUT rangefinders are promising in the relatively short-range (<10 m) detection for mobile use. Similar to other ultrasonic rangefinders, pMUT rangefinders are competitive to optical (light-based) rangefinder systems in relatively short-range detection because the speed of sound is much smaller than that of light, and they do not require high-speed electronic devices and their power consumptions are typically much lower. For instance, the power consumption of an optical rangefinder is greater than 1 W [13], while that of a pMUT rangefinder system can be only a few μW. It has been reported that a pMUT rangefinder can detect over a 1 m range under 16 μW of power consumption [15]. In addition, optical rangefinder systems are easily interfered with by sunlight or surrounding artificial lights, which may heavily affect range accuracy, limiting their use [16].

Secondly, compared to other ultrasonic rangefinders, pMUT rangefinders have their advantages. Currently, most commercially available ultrasonic rangefinders are made of bulk piezoelectric ultrasonic transducers due to their high output power [17], but they suffer from a series of drawbacks, including poor acoustic impedance matching [18] and large size [19]. Although incorporating acoustic matching layers can improve impedance matching [20], the bandwidth is limited, and it may not be practical in specific cases [21,22]. In contrast, pMUTs can have miniaturized size [23,24] and better acoustic matching [22,25], so a higher transmitting efficiency can be achieved. Furthermore, the small size of pMUTs not only enables pMUT rangefinders to be integrated with electronics, which saves space and energy for mobile use but also enables the fabrication of large ultrasonic transducer arrays, helping to realize more complicated systems [26,27,28]. Compared to ultrasonic rangefinders based on capacitive micromachined ultrasonic transducers (cMUTs) [29,30], pMUT rangefinders have a larger ultrasound output, lower bias voltage, and larger detectable range [31,32]. Hence, the low power consumption, relatively high output pressure, and integrated property of pMUT rangefinders make them promising for implementation into mobile devices for range-finding.

pMUT rangefinders were developed after bulk piezoelectric ultrasonic rangefinders, and the last decade witnessed the rapid development of pMUT rangefinders. At first, a single pMUT device with an AlN-based circular membrane was used to form a pMUT rangefinder system in 2010 [33]. After that, pMUT rangefinders with pMUT arrays, different types of pMUT device structures, and piezoelectric materials were developed to enhance the performance of pMUT rangefinder systems [34]. In addition to the advancements of pMUT devices, different range-finding methods are proposed [35]. Furthermore, various derivative systems based on pMUT rangefinders have been demonstrated, such as target positioning and object detection systems [16]. pMUT rangefinders are promising but still relatively new.

The main purpose of this article was to provide a state-of-the-art review of rangefinders based on pMUTs and a future outlook as well. Section 2 explains the principle of pMUT rangefinders and analyzes the performance of pMUT rangefinders. Section 3 reviews how the performance of pMUT rangefinders has improved through the advancement of pMUT devices. Section 4 reviews advanced pMUT rangefinders based on improving the range-finding method. Section 5 briefly introduces derivative systems of pMUT rangefinders. Section 6 summarizes this work and provides future outlooks of pMUT rangefinders.

## 2. Principle of pMUT Rangefinders

The working principles of pMUT rangefinders and the main factors affecting their performance are analyzed in this section to provide a foundation for pMUT rangefinder mechanism analysis and performance evaluation. In Section 2.1, the working mechanism and the signal propagation process of pMUT rangefinders are described. In Section 2.2, the analysis of pMUT rangefinder performance is given, and a quantitative relationship between the range and range error that determines the performance of pMUT rangefinders is deduced. After that, the main factors affecting the performance of pMUT rangefinders are analyzed in Section 2.3.

### 2.1. The Working Mechanism of pMUT Rangefinders

The block diagram of a pMUT rangefinder system is shown in Figure 2, where an electric transmitting signal from a signal generator is converted to an acoustic signal by the transmit mode of a pMUT device (①). The acoustic signal is reflected by the target (②), received by the receive mode of the pMUT device (③), which is converted back to the electric form (④) and picked up by an oscilloscope (⑤). Finally, the target range is given by data processing (⑥).

For traditional pMUT rangefinder systems, a single circular pMUT device is used to transmit and receive signals with a working frequency f, where the waveforms of the transmitting and receiving signals are pulses and echoes, and the system is based on the time-of-flight (TOF) method (also named the pulse-echo method) to find the range of the target. The threshold TOF method is the most commonly used one for pMUT rangefinders due to its simple implementation, where the TOF is measured by the time interval between the starting point and the time at which the leading edge of the echo reaches the amplitude threshold [37], as shown in Figure 3. The range between the pMUT and the target is given by [38]
(1)$$R=\frac{c\cdot \mathrm{TOF}}{2}$$
where c is the sound speed.

Let us look at the signal propagation processes. The acoustic surface pressure amplitude $P_{tx}$ generated by the transmit mode of the pMUT is proportional to the driving voltage $V_{d}$ to excite the transmission mode of the pMUT, i.e.,
(2)$$P_{tx}=G_{t}V_{d}$$
where $G_{t}$ is the transmitting sensitivity of the pMUT. The signal is spread spherically, reflected by the target, and received by the pMUT. For far-field radiation, when acoustic absorption is neglected, the on-axis acoustic pressure function p(r,t) at a distance r from a circular-membrane pMUT and a time, t, is given by [4,40,41]
(3)$$p\left(r,t\right)=j\rho_{ac}cu_{0}\cdot \frac{R_{0}}{r}e^{-j\left(\omega t-kr\right)}$$
where j is the imaginary unit, $\rho_{ac}$ is the mass density of the medium, $u_{0}$ is the average velocity of the membrane, $R_{0}$ is the Rayleigh distance, ω=2πf is the angular frequency, and k is the wave number. In Equation (3), the amplitude of the term “$\rho_{ac}cu_{0}$” is equal to $P_{tx}$ [41]. Equation (3) is usually used to calculate the sound pressure level [4] or the transmitting sensitivity [42] of a pMUT in its transmit mode since $P_{tx}$ in Equation (2) is not directly measurable and must be derived using Equation (3). For the cases in which acoustic absorption is not neglectable (e.g., in-air range-finding) and the acoustic gain $G_{ac}$, which is related to the target geometry [31], must be considered, Equation (3) is changed to:(4)$$p\left(r,t\right)=j\rho_{ac}cu_{0}\cdot \frac{R_{0}}{r}G_{ac}e^{-\alpha r}e^{-j\left(\omega t-kr\right)}$$
where α is the absorption coefficient, which is used to evaluate the energy losses along the acoustic propagation path in the medium. Note that α is dependent on the properties of the medium, including the viscosity, thermal conductivity, mass density, etc., and is positively related to the ultrasound frequency [43].

Due to the round-trip propagation between the pMUT and the target, the amplitude of the received pressure at the pMUT, $P_{rx}$, is obtained by setting r=2R in Equation (4), where *R* is the target range, i.e.,
(5)$$P_{rx}=P_{tx}\frac{R_{0}}{2R}G_{ac}e^{-2\alpha R}$$
The received acoustic signal is converted to the electric signal by the pMUT in the receive mode, and the received voltage is given by:(6)$$V_{rx}=G_{r}P_{rx}$$
where $G_{r}$ is the receiving sensitivity of the pMUT. The signal propagation processes are listed in Table 1 in order.

The relationship between the transmitting and receiving signals is described by Equations (2), (5) and (6), which will be used to analyze the performance of a pMUT rangefinder.

### 2.2. The Performance Evaluation of pMUT Rangefinders

A pMUT rangefinder is used for range-finding, so the maximum measurement range, $R_{max}$, is the main criterion to evaluate its performance. Meanwhile, with the increase in the target range, the received pressure $P_{rx}$ is reduced and $V_{rx}$ is decreased as well, according to Equations (5) and (6), and the accuracy of the range detection and the reliability of the range-finding result is lowered. Therefore, both $R_{max}$ and the range-finding accuracy must be considered for the performance comparison of different pMUT rangefinders.

The range-finding accuracy of a pMUT rangefinder is determined by multiple aspects, including $V_{rx}$, noises and interferences, the bandwidth, and the range-finding method. A large receiving signal amplitude makes the feature points, especially the leading edge, clearer, so the TOF determination and range detection can be more accurate. $V_{rx}$ is related to both the electrical and acoustic energy conversion processes according to Equations (2) and (6). With the increasing distance of the target, $V_{rx}$ or $P_{rx}$ decreases due to two main effects: one is that the signal is spread spherically and only a very small fraction of the energy is reflected by the target and received by the pMUT, and the other is acoustic absorption in the medium, according to Equation (5). Noises and interferences in a pMUT rangefinder system blur the signal and thus reduce the range-finding accuracy. Primary noises and interferences include the thermal noise caused by the random motion of the medium molecules received by the pMUT, the thermal noise of the piezoelectric element, the noises from the receiver circuit, the crosstalk between adjacent pMUT elements, the environmental reflections, and the possible multipath fading if the signal is continuous [44,45,46]. The signal-to-noise ratio (SNR) describes and evaluates the effects of $V_{rx}$ as well as the noise and interference level on the accuracy. Combining Equations (2), (5) and (6) with the definition of SNR produces [37]:(7)$$SNR=\frac{S}{N}=\frac{\frac{1}{2}V_{rx}{}^{2}}{\bar{n^{2}}}=\frac{V_{d}{}^{2}}{2\bar{n^{2}}}\cdot {\left(G_{t}G_{r}G_{ac}R_{0}\right)}^{2}\cdot \frac{e^{-4\alpha R}}{4R^{2}}$$
where S is the signal power, N is the noise power, and $\bar{n^{2}}$ is the mean-square of the noise voltage. Apart from the factors described by SNR, the range-finding accuracy is also affected by the pMUT bandwidth (BW) and the range-finding method. Ideally, the receiving signal is not spread and/or distorted compared to the transmitting signal, and the feature points, such as the leading edge, are sharp. However, due to the narrowband frequency response of the pMUT’s membrane [11], the rise and fall times of the receiving signal depend on the BW, and the feature point positioning error is introduced by the reduction in the signal rise and fall slope. In addition, pMUT rangefinder systems using different range-finding methods have different signal waveforms or the methods to derive the target range from the signals are different, so the accuracies of the ranging results are not the same. To include those multiple aspects influencing the range-finding accuracy, a range error is defined and commonly used for range-finding accuracy evaluation [37]. For the threshold TOF method, which is the most commonly used for pMUT range-finding, the root-mean-square (RMS) range error, $\delta_{R}$, is given by [47]
(8)$$\delta_{R}=\frac{c}{2BW}\frac{1}{{\left(2\cdot SNR\right)}^{\frac{1}{2}}}$$

In this paper, the RMS range error is abbreviated as the range error. The accuracy of a pMUT rangefinder system is revealed by its maximum range error $\delta_{R}{}_{max}$. Overall, in this paper, the maximum range $R_{max}$ and the maximum range error $\delta_{R}{}_{max}$ are mainly considered to evaluate the performance of pMUT rangefinders.

In Equation (8), the bandwidth for a single-mode membrane is given by $BW=\frac{f_{0}}{Q}$, where $f_{0}$ is the resonant frequency and Q is the quality factor [48,49]. Combining Equations (7) and (8) produces:(9)$$\;\delta_{R}=\frac{Q\cdot c\cdot \sqrt{\bar{n^{2}}}}{G_{t}G_{r}G_{ac}R_{0}f_{0}V_{d}}\cdot R\cdot e^{2\alpha R}$$

According to Equations (7) and (9), when R increases, SNR almost exponentially decreases while $\delta_{R}$ almost exponentially increases. This prediction is in good agreement with the experimental results of pMUT rangefinders, as shown in Figure 4. If the range error is large, the range-finding result given by the pMUT rangefinder is inaccurate. Plugging $\delta_{R}{}_{max}$ and $R_{max}$ into Equation (9) yields
(10)$$\;\delta_{R}{}_{max}=\frac{Q\cdot c\cdot \sqrt{\bar{n^{2}}}}{G_{t}G_{r}G_{ac}R_{0}f_{0}V_{d}}\cdot R_{max}\cdot e^{2\alpha R_{max}}$$

When $\delta_{R}$ is required to be less than the maximum allowed range error $\delta_{R}{}_{max}$, the $R_{max}$ that a pMUT rangefinder can achieve is determined by multiple factors that will be discussed in Section 2.3.

### 2.3. Main Factors Affecting pMUT Rangefinder Performance

According to Equation (10) and the previous analysis, with the enhancements of $G_{t}$ and $G_{ac}$, an increased $R_{max}$ can be achieved with fixed $\delta_{R}{}_{max}$, where $G_{ac}$ is related to the target size and the directivity that can be changed by different pMUT structures [50], PCB baffle sizes [51], etc. A commonly used way to increase $G_{ac}$ by making the signal more directional is to add a horn on the pMUT [52]. In addition, with the increase in $G_{r}$, the receiving electric signal amplitude is increased with the side effect that the noise caused by the acoustic thermal noise is also increased. However, since acoustic noise is only one of the noise sources, the SNR is still largely increased with the enhancement of $G_{r}$ and thus $R_{max}$ can be increased accordingly. $G_{t}$, $G_{r}$, and $G_{ac}$ can be enhanced by the pMUT device improvement.

Admittedly, the $R_{max}$ can be increased or the $\delta_{R}{}_{max}$ can be decreased by increasing the value of the driving voltage $V_{d}$ according to Equation (10). However, higher driving voltage means higher power consumption and even nonlinearity [13], so both high performance and a moderate driving voltage should be achieved. In addition, the $R_{max}$ is positively related to the Rayleigh distance $R_{0}$ with a fixed $\delta_{R}{}_{max}$ in Equation (10), and $R_{0}$ is given by [40]:(11)$$R_{0}=\frac{A}{\lambda }=\frac{fA}{c}$$
where A is the membrane surface area, λ is the wavelength, and f is the frequency of the signal transmitted by the pMUT and is referred to as the working frequency in this paper. According to the above equation, increasing A leads to a higher Rayleigh distance $R_{0}$, and thus a larger $R_{max}$ with fixed $\delta_{R}{}_{max}$, but the size and power consumption of the pMUT are also increased, and the resonant frequency is changed as well [53]. Therefore, the strategy to enhance the performance by directly changing the membrane area is rarely adopted.

Some factors have dual effects on the performance that should be properly designed, including the quality factor Q. Firstly, the increase in Q causes a decrease in the bandwidth, so $R_{max}$ is decreased or $\delta_{R}{}_{max}$ is increased, according to Equation (10). Secondly, the transmitting and receiving sensitivities $G_{t}$ and $G_{r}$ are also relevant to Q. A high Q means low acoustic and structural damping of pMUTs [54]. Therefore, with the increase in Q, the energy conversion efficiency is increased, and $G_{t}$ and $G_{r}$ are enhanced. Because of the dual effects of Q on the parameters of pMUTs, Q should be designed to be a moderate value to balance the high bandwidth and high sensitivity requirements for lower range error and larger detection range.

The working frequency f and the resonant frequency $f_{0}$ also have multiple effects on pMUT rangefinder performance. f is the actual frequency of the pMUT driving signal while $f_{0}$ is a property of the pMUT and is determined by the pMUT material and structure. In most pMUT rangefinder systems, to ensure the maximum vibration of the pMUT membrane, f should be set as approximately equal to $f_{0}$ [15]. The signal amplitude reduction due to the difference between f and $f_{0}$ causes the increase in $\delta_{R}{}_{max}$ or decrease in $R_{max}$. According to Equation (10), for media with high absorption, the primary effect of f is acoustic absorption. An example of the relationship between the transmission loss and the range in a spread-only situation without absorption and in acoustic absorption situations with different frequencies is shown in Figure 5. With the decrease in f, the absorption coefficient α is decreased [40], so the transmission loss is reduced, and the $R_{max}$ is increased with fixed $\delta_{R}{}_{max}$. In addition, with the reduction in f, the electric and mechano-acoustic impedances are decreased, so the receiving sensitivity $G_{r}$ can be enhanced [54]. However, f should not be too low because the Rayleigh distance $R_{0}$ is decreased with f reduction according to Equation (11), resulting in a higher range of error. Furthermore, since $f_{0}$ should be close to f, with small f, the pMUT device has to be redesigned to have lower $f_{0}$, which in turn causes lower bandwidth and higher range error. In addition, the quality factor Q may be also altered with the change in $f_{0}$ [55]. Therefore, multiple effects of f and $f_{0}$ must be considered during the design of the pMUT in order to obtain the optimal frequency of the pMUT rangefinder system in different scenarios. For the in-air range-finding, the working frequency should be moderately low due to the large absorption of air. For instance, a pMUT rangefinder first achieving more than a five-meter detection range with over 11.5 dB SNR was proposed by Yang et al. [31], where the working frequency was 66 kHz, which is lower than most previous designs. For in-liquid range-finding, the acoustic absorption is low, and the working frequency is usually in the MHz range [46] for higher bandwidth and larger Rayleigh distance.

The mean-square of the noise voltage $\bar{n^{2}}$ and the absorption coefficient α negatively affect pMUT rangefinder performance. $\bar{v_{n}{}^{2}}$ is determined by the noise and interference sources from the pMUT rangefinder and environment that are discussed in Section 2.2. α is determined by the working frequency f and properties of the medium that are discussed in Section 2.1. The reduction in $\bar{n^{2}}$ and/or α gives rise to higher $R_{max}$ or lower $\delta_{R}{}_{max}$, according to Equation (10).

Overall, Table 2 lists the main factors affecting pMUT rangefinder performance and the trends of the maximum range $R_{max}$ with the increase in those factors when the maximum range error $\delta_{R}{}_{max}$ is fixed. Some of the factors could be improved by employing advanced pMUT devices and range-finding methods. Strategies to improve pMUT devices and range-finding methods for high-performance pMUT rangefinders are analyzed in detail in Section 3 and Section 4, respectively.

## 3. Advancements of pMUT Rangefinders Based on Improvements of pMUT Devices

A typical pMUT rangefinder system is made of a single pMUT device with a circular membrane and uses AlN as the piezoelectric material. This type of pMUT rangefinder has limited performance. In order to achieve the higher performance pMUT rangefinder systems, pMUT arrays, advanced pMUT device structures, and novel piezoelectric materials have been applied to pMUT rangefinders, which are discussed in detail in this section.

### 3.1. pMUT Rangefinders with pMUT Arrays

Using pMUT arrays to substitute a single pMUT for range-finding can help to achieve a higher maximum range or lower range error. As more pMUT elements are used for signal transmitting and receiving, the total transmitting and receiving acoustic signal amplitude are manyfold increased, so the SNR is increased, and the range error is largely reduced [13]. In addition, pMUT arrays with more element numbers can help to improve the directivity [56,57], so more acoustic energy is transmitted along the direction of the target and thus the receiving signal is enhanced, which in turn improves the maximum range and reduces the range error. In addition, since less energy along other directions is spread, the environmental reflection that affects the noise level is reduced, and thus $\bar{n^{2}}$ is lowered, as discussed in Section 2.2. For a one-dimensional (1D) array, assuming that the directivity with a single pMUT element is one, the directivity function D(θ) with an array of pMUT elements is given by [58]
(12)$$D\left(\theta \right)=\frac{\sin\left(\frac{\pi Nd}{\lambda }\sin\theta \right)}{N\cdot \sin\left(\frac{\pi d}{\lambda }\sin\theta \right)}$$
where N is the number of elements, d is the spacing between adjacent elements, and θ is the azimuth. The directivity function is represented by the normalized sound pressure and is shown in Figure 6. According to Figure 6, with the increase in the element number, the width of the main lobe is decreased, which means that the directivity is improved. The theory is also valid for two-dimensional (2D) arrays [59].

The types of pMUT array structures include line arrays (1D arrays) [60,61], polygon (e.g., hexagon) arrays [62,63], and square arrays [55,64], as shown in Figure 7. In many rangefinder systems using pMUT arrays, in order to keep the transmitter electrodes and the receiver electrodes in the same vibrational mode, respectively, all the transmitter electrodes are connected by metal bridges, and all the receiver electrodes are connected as well [65,66].

Advanced pMUT array designs can improve the factors affecting pMUT performance, in which the density of pMUT elements and the mechanical or acoustic crosstalk between pMUT elements must be considered. By increasing the density of pMUT elements, the transmitting and receiving signal amplitude per unit area is enhanced. However, when reducing the spacing between pMUT elements to increase the element density, the crosstalk or cross-coupling between different elements must be carefully considered [23]. Strategies for minimizing the pMUT array crosstalk have been developed. For example, Yang et al. analyzed the mutual-coupling effect with different spacings in a square array of pMUTs and figured out the optimal spacing under a certain working frequency [23]. Moreover, Xu et al. proposed a resonant cavity design for a pMUT array to reduce crosstalk [67]. Vysotskyi et al. overviewed and compared acoustic isolation techniques for pMUT arrays concerning crosstalk reduction [68].

Table 3 lists the parameters of pMUT rangefinders with a single pMUT or different pMUT arrays. According to Table 3, pMUT arrays with large numbers of elements generally have high maximum ranges or high accuracies, which accords with the analysis. Note that the examples listed in Table 3 have different structures, piezoelectric materials, driving voltages, and $\delta_{R}{}_{max}$ or $SNR_{min}$. Thus, more control experiments of range-finding using single pMUTs and pMUT arrays with different element numbers need to be carried out in the future to further evaluate the contribution of pMUT arrays on the performance enhancement of pMUT rangefinder systems.

### 3.2. pMUT Rangefinders with Advanced pMUT Device Structures

In addition to using pMUT arrays to enhance range-finding performance, pMUT rangefinders with advanced device structures have been exploited as well. For instance, some key parameters, including the transmitting and receiving sensitivities, are improved by optimizing the membrane shape of pMUT devices. Most conventional pMUT rangefinders use circular membrane pMUTs; nonetheless, some pMUT rangefinder designs have adopted square membranes for pMUTs recently, as shown in Figure 8. The vibrational modes of a pMUT are determined by the shape of the piezoelectric membrane, as shown in Figure 9. For the same device size, square-membrane pMUT units have a higher coverage area than circular ones [42], so the membrane vibration displacement of square-membrane pMUTs is largely higher than circular ones with the same driving voltage, which is proved by a comparison experiment in [71], as shown in Figure 10. Therefore, square membrane pMUT rangefinders have the potential for higher sensitivities $G_{t}$ and $G_{r}$ than current circular ones and achieve a larger detection range and lower range error. Chiu et al. reported a square-membrane pMUT rangefinder system [72] that achieved a higher maximum range (0.5 m) and lower range error (0.63 mm) than a comparable circular-membrane pMUT rangefinder (0.45 m maximum range with 1.3 mm error) [33]. In the future, the square-membrane pMUT structure may be combined with other strategies to further enhance pMUT rangefinder performance.

Apart from the membrane shape change, Zhou et al. proposed a dual-electrode pMUT rangefinder system [15], as shown in Figure 11a. The pMUT transmitter has both inner and outer electrodes, and the driving electric signals of the electrodes have a 180°-phase difference. The electrodes with anti-phase driving signals control the displacement field of the membrane and improve the transmitting sensitivity. Therefore, the output acoustic pressure of the dual-electrode mode transmitter with anti-phase driving signals is considerably higher than the single electrode one, as shown in Figure 11b. Range-finding experiments were carried out, and the rangefinder system in the dual-electrode mode exhibited a 2.2 m maximum distance (i.e., a 1.1 m maximum range in the reflection target detection), which is 29.4% larger than the single electrode mode.

There are other advanced pMUT device structures that are promising to be used for high-performance range-finding. For example, a bimorph structure is an effective way for pMUTs to enhance sensitivities, where the membrane of a pMUT contains two piezoelectric layers [73,74]. Because the number of piezoelectric layers is doubled, the vibration displacement and acoustic pressure of bimorph pMUTs are much larger than those of unimorph ones [75]. Shao et al. utilized a bimorph structure and developed a pMUT rangefinder derivative system to detect a target over 1 m away with a more than ±65° field of view [76], which was greater than the ±45° field of view of a unimorph design [13]. Moreover, other advanced structure designs, including the venting structure [50], dome shape [77], and unclosed membrane [78], may be further explored in order to extend the maximum range and lower the range error of pMUT rangefinders in the future.

Table 4 lists the parameters of typical pMUT rangefinders with different structures found in the literature. The maximum ranges and range errors of these pMUT rangefinders will also be illustrated in Figure 12.

### 3.3. pMUT Rangefinders with Novel Piezoelectric Materials

The transmitting sensitivity $G_{t}$ and the receiving sensitivity $G_{r}$ are mainly dependent on the piezoelectric materials of pMUT devices [79]:(13)$$\{\begin{array}{c} G_{t}\propto \left|e_{31,f}\right|\\ G_{r}\propto \left|\frac{e_{31,f}}{\varepsilon_{0}\varepsilon_{33}}\right| \end{array}$$
where $e_{31,f}$ is the piezoelectric constant, $\varepsilon_{0}$ is the vacuum permittivity, and $\varepsilon_{33}$ is the relative dielectric constant in the z-direction. Therefore, the product of the sensitivities $G_{t}G_{r}$ can be described by $\frac{e_{31,f}{}^{2}}{\varepsilon_{0}\varepsilon_{33}}$, which is denoted as the figure of merit for a pMUT [80]. Since larger $G_{t}G_{r}$ give rise to a smaller maximum range error $\delta_{R}{}_{max}$ or larger maximum range $R_{max}$ according to Equation (10), researchers have been working on choosing different piezoelectric materials and strategies to enhance $G_{t}G_{r}$ by increasing the absolute value of $e_{31,f}$ or decreasing $\varepsilon_{33}$ in order to enlarge the maximum range of pMUT rangefinders [81,82,83].

ZnO [84], PZT [85,86,87], and AlN [88,89,90] are widely used materials for pMUT devices with the advancement of thin-film deposition technologies. The drawback of ZnO is that it has a relatively high conductivity [91]. A PZT-based pMUT rangefinder has higher $G_{t}$ due to its high piezoelectric coefficients, but its $G_{r}$ is limited because of its high dielectric constants, so the $G_{t}G_{r}$ or $\frac{e_{31,f}{}^{2}}{\varepsilon_{0}\varepsilon_{33}}$ of PZT is not largely higher compared to that of AlN, as shown in Table 5 [92,93]. Furthermore, AlN is more compatible with CMOS fabrication than ZnO and PZT, which enables AlN pMUT rangefinders to be integrated in chips [92,94]. Overall, AlN is the most commonly used material for pMUT rangefinders.

However, AlN-based pMUT rangefinders have limited performance due to their low piezoelectric coefficients $e_{31,f}$ and high sensitivity to parasitic capacitance [91]. Since conventional PZT-based pMUTs cannot replace AlN-based pMUT rangefinders for their poor $G_{r},$ as it has been explained above, researchers applied various strategies to improve PZT films for advanced pMUT rangefinders with performance surpassing current AlN-based ones [95]. For example, S. Yoshida et al. developed a monocrystalline Pb(Mn_1/3_,Nb_2/3_)O_3_-PZT (PMnN-PZT) epitaxial thin film pMUT rangefinder [15]. According to Table 5, PMnN-PZT has a higher $\frac{e_{31,f}{}^{2}}{\varepsilon_{0}\varepsilon_{33}}$, so it has a large $G_{t}G_{r}$. Therefore, the maximum range of PMnN-PZT pMUT rangefinders is increased to 2.2 m, i.e., 1.1 m in the reflection mode. Another example is the design of the single-crystal-PZT pMUT rangefinder developed by Luo et al. in 2020 [70], which has a maximum range of 2.4 m in the reflection mode. The $\frac{e_{31,f}{}^{2}}{\varepsilon_{0}\varepsilon_{33}}$ of single-crystal PZT is not only obviously higher than conventional PZT but is also more than that of PMnN-PZT, according to Table 5. Furthermore, the single-crystal-PZT pMUT rangefinder in [70] has lower resonant frequencies (40~50 kHz) than the PMnN-PZT (>150 kHz) rangefinder in [15]. Since moderately low resonant frequencies are favorable for in-air pMUT rangefinders with long-range detection according to Section 2.3, the single-crystal-PZT-based pMUT rangefinder in [70] achieves higher $R_{max}$.

**Table 5 micromachines-14-00374-t005:** Comparison of piezoelectric materials for pMUT rangefinders. Reproduced with permission from [70], Copyright 2021, IEEE.

Materials	$e_{31,f}$ (C⋅m^2^)	$\varepsilon_{33}$	$\frac{e_{31,f}{}^{2}}{\varepsilon_{0}\varepsilon_{33}}$ (Gpa)
AlN [96,97]	−1	10	10.8
PZT (2 μm) [27]	−13.1	854	22.7
PMnN-PZT [15]	−14	~250	88.6
Single-crystal PZT [70]	−16~−24	308	93.7~211
20% ScAlN [98,99]	−1.6	12	24.1

Apart from totally changing the piezoelectric material from AlN to other ones, other research has focused on doping AlN with suitable materials such as Scandium (Sc) to achieve a high piezoelectric coefficient [100,101]. This strategy balances the objectives that keep the advantages of AlN and increase the absolute value of $e_{31,f}$. Yang et al. demonstrated a pMUT rangefinder array based on AlScN film, and a long-range detection with a maximum range of 6.8 m and SNR ≥ 11.5 dB was achieved [31].

Table 6 lists the parameters of pMUT rangefinders with different piezoelectric materials. According to Table 6, pMUT rangefinders based on PMnN-PZT, single-crystal PZT, and AlScN were observed to have obviously higher detection ranges than those on AlN and PZT. In particular, the AlScN pMUT rangefinder reaches the largest detection range (6.8 m), and the PMnN-PZT pMUT rangefinder in [15] realizes a long-range measurement of 1.1 m with very low power consumption (0.5 V driving voltage and about 16 μW), while a counterpart AlN pMUT rangefinder needs 10 V to reach the comparable $R_{max}$.

The performances of TOF pMUT rangefinders with different pMUT devices are collectively presented in Figure 12. Note that for the performance comparison between two individual rangefinders, both $R_{max}$ and $\delta_{R}{}_{max}$ must be considered at the same time. For instance, the one with the same $\delta_{R}{}_{max}$ but higher $R_{max}$, or the one with the same $R_{max}$ but lower $\delta_{r}{}_{max}$, or the one with a higher $R_{max}$ and lower $\delta_{R}{}_{max}$ has a higher performance. In the figure, the colors of the markers stand for the working frequency f (f is substituted by the resonant frequency $f_{0}$ for those pMUT rangefinders that did not explicitly provide f, and f is averaged for those with multiple working frequencies), ○ is a circular membrane pMUT, and ◊ is a square membrane pMUT. The hollow markers 🞉 and 
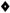
 refer to those with no pMUT arrays used, while the solid markers ● and ⧫ refer to those with pMUT arrays as transmitters and/or receivers. Those pMUT rangefinders that used piezoelectric materials other than AlN are labeled. It can be seen in Figure 12 that moderately low working frequencies are desirable for pMUT rangefinders to achieve long-range detection, and the suitable working frequency range is approximately 40~200 kHz. Employing piezoelectric materials with high piezoelectric constants in pMUTs is the most effective means of extending the maximum range. In addition, pMUT arrays also contribute to the range error reduction or the maximum range enlargement. Square-membrane pMUT rangefinders have larger maximum ranges than their circular membrane counterparts, although the number of reported square-membrane pMUT rangefinders is relatively small.

**Figure 12 micromachines-14-00374-f012:**
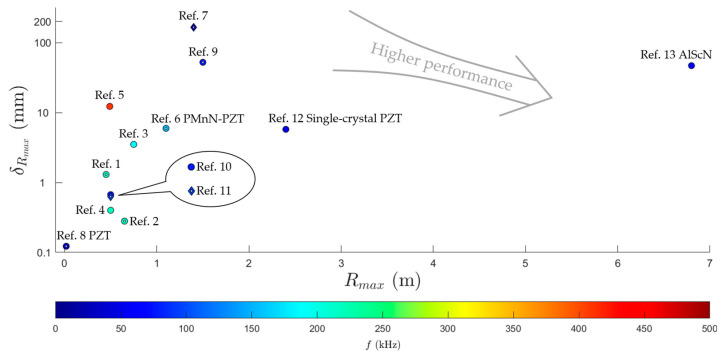
Performance comparison illustration of different TOF pMUT rangefinders. All the ranges and the corresponding range errors that are measured by the distance between the transmitter and the receiver are divided by two to be converted to the reflection mode results for performance comparison. 3σ errors are divided by three to be converted to the RMS errors. For those pMUT rangefinders that did not report their maximum range errors, $\delta_{R}{}_{max}$ is estimated by Equation (8), where the sound speed is set to be c= 343 m/s according to [103]. Data are from Ref. 1 ([33]), Ref. 2 ([11]), Ref. 3 ([52]), Ref. 4 ([13]), Ref. 5 ([55]), Ref. 6 ([15]), Ref. 7 ([36]), Ref. 8 ([102]), Ref. 9 ([39]), Ref. 10 ([62]), Ref. 11 ([72]), Ref. 12 ([70]), and Ref. 13 ([31]) in published year sequence.

## 4. Advancements of pMUT Rangefinders Based on Improvements of Range-Finding Methods

Section 3 describes the ways to enhance the performance of pMUT rangefinder systems by improving pMUT devices. Another form of performance enhancement is the improvement of the range-finding method. pMUT range-finding systems based on advanced TOF range-finding methods achieve high accuracy and wide detection range in long-range detection and are analyzed in Section 4.1. pMUT range-finding systems based on continuous wave (CW) range-finding methods achieve high accuracy in short-range detection and are analyzed in Section 4.2.

### 4.1. Advanced Time-of-Flight Range-Finding Methods for pMUT Rangefinders

#### 4.1.1. TOF with Cross Correlation for Enhanced Accuracy

According to Section 2.1, traditional TOF pMUT rangefinder systems only use a single point to calculate the range, which may have a large range error because the attenuated received signals combined with considerable noises and interferences obscure the position of the leading edge. To address this problem, a cross-correlation TOF system was proposed, where signals were sampled, cross-correlation was performed, and the whole envelopes of the transmitting and receiving signals were compared and calibrated to determine the time of flight. Sampling sequences of the transmitting and receiving signals were cross-correlated [104], and the lag of the maximum cross-correlation peak, $\tau_{max}$, was given. Then, the distance between the transmitter and receiver, L, was given by [19,105]
(14)$$L=\left(\frac{\tau_{max}}{f_{s}}-\mathrm{TOE}\right)c-L_{cal}$$
where $f_{s}$ is the sampling frequency, TOE is the time of the emission of the signal, and $L_{cal}$ is a calibration constant. The range R between the pMUT rangefinder and the target can be obtained by combining Equation (14) with R=L/2.

In the cross-correlation method, the range result is determined not only by one point but by multiple feature points on the envelope, so the random noise can be mitigated and $\bar{n^{2}},$ in Equation (10), can be decreased, resulting in a lower range error and higher accuracy. Furthermore, in order to diminish the envelope distortion during energy conversion processes, the envelope shape of the transmitting signal in this system can be well designed to have enough smooth rising and falling slopes, as well as containing enough feature points [33]. A practical design of the envelope shape and the diagram of a pMUT rangefinder system applying the cross-correlation TOF method is shown in Figure 13 [33].

R. Przybyla et al. developed the first pMUT rangefinder system based on the cross-correlation TOF method as early as 2010 [33]. Many recent works also use cross correlation to obtain the detection range [31,106].

#### 4.1.2. TOF with Ring-Down Suppression for Lower Blind Area

The maximum range is often the main parameter to be enhanced in most research on pMUT rangefinders in order to make pMUT rangefinders more competitive and to be applied to commercial range-finding use. Nevertheless, some recent research demonstrated strategies to lower the minimum detectable range. TOF pMUT rangefinder systems with a single pMUT device have a relatively large minimum detectable range due to the ring-down effect of the vibrating membrane, as shown in Figure 14. The ring-down effect is the main cause to extend the blind area of a pMUT rangefinder because the signal received during the ring-down stage is overlapped or distorted by the ring-down vibration, where the range error is largely increased, and the range-finding may fail [107]. The minimum detectable range of a TOF pMUT rangefinder can be defined by the blind area duration.

A simple way to avoid the blind area is to use two pMUT devices that are physically separated, where one of the pMUTs always works at the TX mode and the other always at the RX mode. However, the device size, readout circuit, and power consumption are largely increased if doubling the number of pMUT devices. Another strategy to lower the minimum detectable range in relatively short-range detection for pMUT rangefinders working in switching TX and RX modes is to redesign the transmitting signal to suppress the ring-down effect and reduce the blind area. For instance, Pala et al. proposed a method of ring-down suppression [108], in which the transmitting signal consisted of a number ($N_{p}$) of normal cycles, such as the excitation signal and a number ($N_{n}$) of cycles that are 180° phase-shifted as the suppression signal, as shown in Figure 15a. The normal cycle and the 180° phase-shifted cycle signals counteract the vibration of the pMUT membrane and accelerate the vibration attenuation so that the ring-down time is reduced. The number of cycles $N_{p}$ and $N_{n}$ must be properly determined. When $N_{n}$ is not large enough, the counteracting effect is not adequate, while when $N_{n}$ is too high, the additional 180° phase-shifted signal over-acts and excites the vibration of the pMUT membrane in its phase and frequency. In the experiment of [108], the optimal choice is $N_{p}=10$ and $N_{n}=6$, as shown in Figure 15b, and the minimum detectable range (the blind area) is reduced from 13.91 cm to 12.16 cm. Furthermore, Wu et al. demonstrated a transfer function-based ring-down suppression method [107], which also achieved ring-down reduction through excitation signal suppression, but the suppression signal waveform was improved, as shown in Figure 15c, where the suppression signal was generated by a system including a transfer function and a simple proportion controller. The comparison experiment was carried out, and the minimum range (the blind area) was reduced by about 40% [107], as shown in Figure 15d. In the future, accuracy experiments of the ring-down suppression methods should be carried out to obtain the range error or SNR of the pMUT rangefinders using the methods in order to evaluate the accuracy of those methods in relatively short range detection compared with other designs.

### 4.2. Advanced Continuous Wave Range-Finding Methods for pMUT Rangefinders

#### 4.2.1. MFCW pMUT Rangefinder System

The accuracy of TOF-based pMUT rangefinders is largely limited by the receiving signal amplitude which is heavily reduced by acoustic absorption [109]. Different from TOF methods that obtain the target range by amplitude information, continuous wave (CW) methods obtain the range by phase difference information, so they can achieve a higher accuracy than TOF methods [35]. Unlike TOF methods that the signals are pulses and echoes, CW methods transmit and receive continuous wave signals to find the range. The vibration modes of the signals in CW methods are steady, so they are less affected by bandwidth limitations. The change in the range-finding method from TOF methods to CW methods changes the relationship between $\delta_{R}$ and SNR [47], so the $\delta_{R}$ can be reduced even if $V_{d}$ and $\bar{n^{2}}$ are not changed.

The simplest CW-based pMUT rangefinder system is to only use a sinusoidal wave as the signal. The phase detector can detect the phase difference between the transmitting and receiving signals, Δφ. The range is half of the distance L between the transmitter and the receiver and is given by [110]:(15)$$R=\frac{L}{2}=\frac{c\Delta \varphi }{4\pi f}$$
where f is the frequency of the signal. However, since Δφ≤2π, $R\leq \frac{c}{2f_{0}}$, so $R_{max}=\frac{c}{2f_{0}}$, which is very small. If the range is more than $\frac{c}{2f_{0}}$, phase ambiguity will occur. The simplest CW-based system cannot be used for pMUT rangefinders due to their low detection range. To address the problem of the detection range limitation, two frequency continuous wave (TFCW) systems have been proposed, where the signal contains two sinusoidal waves with slightly different frequencies $f_{1}$ and $f_{2}$ ($\left|f_{1}-f_{2}\right|=\Delta f$), so the range is given by [111]:(16)$$R=\frac{L}{2}=\frac{c\Delta \varphi }{4\pi \Delta f}$$

Since Δφ≤2π, $R_{max}=\frac{c}{2\Delta f}$, so $R_{max}$ is increased. The range error of a TFCW system is given by [47]:(17)$$\delta_{R}=\frac{c}{2\pi \cdot BW}\frac{1}{{\left(2\cdot SNR\right)}^{\frac{1}{2}}}$$

By comparing Equations (8) and (17), the range error of a TFCW system is π times lower than a threshold TOF system with the same SNR, bandwidth, and sound speed. However, there is a trade-off between $R_{max}$ and $\delta_{R}{}_{max}$ for the TFCW system. Although $\delta_{R}$ is decreased with the increase in Δf, $R_{max}$ is shortened when Δf is increased. To balance the parameters, $R_{max}$ cannot be too high and $\delta_{R}{}_{max}$ cannot be too low.

To solve this problem, multi-frequency continuous wave (MFCW) pMUT rangefinder systems are proposed, where three sinusoidal waves with frequencies of $f_{1}$, $f_{2,}$ and $f_{3}$ are transmitted simultaneously ($f_{1}>f_{2}>f_{3}$). Define that $\Delta f_{i,j}=f_{i}-f_{j}$ is the frequency difference between the ith and jth waves, $\varphi_{i}$ is the phase shift of the ith wave when it arrives at the receiver, and $\Delta \varphi_{i,j}=\varphi_{i}-\varphi_{j}$ is the phase shift difference between the ith and jth waves. The working mechanism of the MFCW method is illustrated in Figure 16, and the range R is given by [111]:(18)$$R=\frac{L}{2}=\frac{c}{2}\left(\mathrm{Int}\left[\frac{\Delta \varphi_{1,2}\cdot \Delta f_{1,3}}{2\pi \Delta f_{1,2}}\right]\frac{1}{\Delta f_{1,3}}+\mathrm{Int}\left[\frac{\Delta \varphi_{1,3}\cdot f_{1}}{2\pi \Delta f_{1,3}}\right]\frac{1}{f_{1}}+\frac{\varphi_{1}}{2\pi f_{1}}\right)$$

As long as $\Delta f_{1,3}$ is sufficiently larger than $\Delta f_{1,2}$, $R_{max}$ can be nearly independently increased with the decrease in $\Delta f_{1,2}$, and the range error can be independently decreased with the increase in $\Delta f_{1,3}$ and $f_{1}$, where $c/\Delta f_{1,3}$ determines the first-order resolution and $c/f_{1}$ determines the second-order resolution. Thus, $R_{max}$ and $\delta_{R}{}_{max}$ are separately controlled [35]. As phenomenologically illustrated in Figure 17, for a TFCW system, when the period of the envelope is extended from Figure 17(a1) to Figure 17(a2), $R_{max}$ increases at the cost that the average slope of the envelope is also reduced, meaning that the resolution and the range error are decreased. In contrast, for an MFCW system, with the increase in the envelope’s period from Figure 17(b1) to Figure 17(b2), $R_{max}$ extends, and the average slope of the envelope (the spines with different heights) nearly does not change, which means that $\delta_{R}{}_{max}$ is approximately not increased. Furthermore, since MFCW systems use phase, not time-of-flight, to calculate the range, they are less affected by the large amplitude attenuation of the signal, which further enhances the accuracy of MFCW pMUT rangefinder systems.

In 2019, Chen et al. developed an MFCW-based pMUT rangefinder system and achieved a less than 0.0711 mm 3σ error (i.e., $\delta_{R}<$ 0.024 mm) when R< 0.1 m (for most TOF-based pMUT rangefinder systems, $\delta_{R}$ are not less than 0.4 mm) [35]. Although the detection range of this MFCW-based system is relatively low due to the maximum range limitation, MFCW-based pMUT rangefinder systems are promising to be used for high-accuracy short-range detection. The block diagram of an MFCW pMUT rangefinder system is shown in Figure 18. Different from a TOF system, the waveform generator must output three continuous waves with different frequencies [35].

Table 7 lists the typical rangefinders with continuous wave methods. According to Table 7, MFCW rangefinders have larger $R_{max}$ and lower $\delta_{R}{}_{max}$ than the TFCW one. The conventional transducer-based range finder has a higher $R_{max}$ than the pMUT-based one, which means that the performance of the pMUT rangefinder should further be improved in the future to be competitive for commercial use.

#### 4.2.2. MFPW pMUT Rangefinder System

MFCW-based pMUT rangefinder systems cannot achieve ideally high performance due to multipath reflections that are caused by the standing waves formed between the pMUT device and the target [45]. To eliminate this side effect, Zamora et al. proposed a novel system configuration for pMUT rangefinders named a multi-frequency pulsed wave (MFPW) system in 2021 [46], where three continuous waves are replaced by three pulses, each of which contains a single sinusoidal wave with several cycles. Additionally, the number of cycles, as well as the interval between distinct pulses, are well designed to make different pulses distinguishable and have a time-of-flight that is more than the total length of the transmitted signal, as shown in Figure 19. $\varphi_{1}$, $\varphi_{2},$ and $\varphi_{3}$ are measured from these three pulses, respectively, and then $\Delta \varphi_{1,2}$ and $\Delta \varphi_{1,3}$ can be determined. The following steps to derive the range are similar as in the MFCW system described above. An MFPW system has a similar calculation process as an MFCW system, but it avoids interferences caused by multipath fading because discontinuous signals cannot form standing waves. The pMUT rangefinder system in [46] has an extremely low error: lower than 6.2 μm within a 3.5 mm transmitter-receiver distance.

MFCW- and MFPW-based pMUT rangefinder systems are suitable to measure a short range (<several decimeters) because the $R_{max}$ is limited by the maximum phase shift difference of 2π. However, MFCW- and MFPW-based pMUT rangefinder systems have shown their extremely low range error when the range is short. Thus, they are competitive for short-range detection with high accuracy.

Overall, there are various kinds of pMUT rangefinder systems that are used for different applications. For long-range detection (>1 m), pMUT rangefinder systems should have moderately low working frequency according to Section 2.3 and utilize TOF range-finding methods. For short-range detection (<0.1 m) with the low-range-error requirement, pMUT rangefinder systems should utilize MFCW or MFPW range-finding methods.

## 5. Derivative Systems of pMUT Rangefinders

pMUT rangefinders described in Section 2, Section 3 and Section 4 can only find the range of a target but not the angle and direction of the target. Nevertheless, a group of pMUT rangefinders can be combined as an array to accomplish more complicated tasks such as target positioning (Section 5.1) and object detection (Section 5.2). These systems are named derivative systems of pMUT rangefinders because they are not simply pMUT rangefinder systems but are extensions of pMUT rangefinder systems.

### 5.1. pMUT Rangefinder Derivative Systems for Target Positioning

A pMUT rangefinder target positioning system can determine the position of a single point in a space so that the range and the angles of the point in a spherical coordinate frame can be measured. A target positioning system consists of N×N pMUT elements that are independently wired, and the time-of-flight of each element, $\tau_{ij}$, is separately calculated. For a target with the azimuths of θ and φ, the differences in the time-of-flight between neighboring elements of the pMUT array are given by [52]
(19)$$\{\begin{array}{c} \Delta \tau_{x}=\frac{d}{c}sin\theta \cdot sin\varphi \\ \Delta \tau_{y}=\frac{d}{c}sin\theta \cdot cos\varphi \end{array}$$
where d is the adjacent element spacing (R≫Nd), $\Delta \tau_{x}$ is the difference in the time-of-flight between neighboring elements along the x-axis, and $\Delta \tau_{y}$ is that along the y-axis. d has to be designed to be smaller than λ/2 to avoid grating lobes [113]. The target range R can be directly obtained by the $\tau_{ij}$ data through Equation (1), and the azimuths θ and φ can be obtained by the known $\Delta \tau_{x}$ and $\Delta \tau_{y}$ data through Equation (19). Finally, the position of the target is given in Figure 20.

### 5.2. pMUT Rangefinder Derivative Systems for Object Detection

pMUT rangefinder arrays can be utilized for object detection as well, where a group of range data of the points of an object [76] or a hand [16] is measured by the pMUT rangefinder array before data processing is used to detect the presence of the object or recognize the object features. An example of a pMUT rangefinder object detection system for gesture recognition is shown in Figure 21a.

Because of the narrowband property of pMUTs, all the elements in a pMUT rangefinder array for object detection have the same working frequency. A CMOS circuitry capable of demodulation, frequency tracking, pulse generation, beamforming, and target tracking should be integrated with a pMUT rangefinder array to construct the system [13], as shown in Figure 21b, where beamforming is the most crucial part. Without deliberate beamforming, the echo signals reflected by multiple points of a target or multiple targets mix and superpose together, so one cannot distinguish a range difference from an angular difference by analyzing the signal differences of distinct elements for a pMUT rangefinder array. Hence, beam steering must be added so that the pMUT array only detects the range of a point in a certain direction at a time and then scans the direction [114,115]. The illustration of steering based on beamforming is shown in Figure 22. When measuring the range of target 1, phase delays are set to the elements of the pMUT array to compensate for the difference in the time-of-flight in the direction of target 1 ($\theta_{1}$), and then the received signals are summed together. Consequently, the received signal from $\theta_{1}$ is intensified, but other signals (e.g., in the direction of target 2 $\left(\theta_{2}\right)$) are not, so the range obtained this time is the range of target 1.

## 6. Summary and Future Outlooks

This article discusses piezoelectric micromachined ultrasonic transducers (pMUTs) for range-finding. A brief introduction of pMUT rangefinders has been given, including their basic structures, working principles, advantages, and applications. Various designs for high-performance pMUT rangefinders have been discussed. pMUT arrays as well as advanced device structures (e.g., bimorph structure) and piezoelectric materials (e.g., AlScN) help to achieve high performance. Additionally, advanced range-finding methods (e.g., the MFPW method) are able to enhance the range-finding accuracy as well. Moreover, derivative systems of pMUT rangefinders that extend the use of pMUT rangefinders are briefly introduced.

With the maturing of advanced pMUT devices, pMUT rangefinders are promising as replacements for bulk piezoelectric transducers for commercial mobile range-finding because of their small size, low cost, and low power consumption. The performance of pMUT rangefinder systems should be further improved to enable pMUT rangefinders to be competitive consumer devices. Specifically, two main aspects should be focused on in the future: (1) the narrowband property of pMUTs makes their bandwidth generally lower than their counterparts, especially cMUTs, so electric and acoustic signal conversion processes inevitably have heavy distortions, and pMUT rangefinders are not competent to high-resolution 3D range-finding detection, especially in the air. The structural design, materials, and fabrication techniques of pMUTs have to be improved to increase their bandwidth, which, unfortunately, has not been extensively researched now. (2) It is still challenging to enable pMUT rangefinders with small sizes to generate a high output pressure that is comparable to bulk rangefinders. Therefore, in the future, pMUT rangefinders should be continuously improved through a deeper understanding of their principles, innovative structures, and/or new materials to reach a comparable output pressure as bulk rangefinders. Furthermore, high-performance range-finding methods such as frequency modulation and binary modulation that have not been applied to pMUT rangefinders are prospective to be researched for pMUT rangefinder systems for a particular use. In conclusion, the future development of pMUT rangefinders is challenging but promising.

## Figures and Tables

**Figure 1 micromachines-14-00374-f001:**
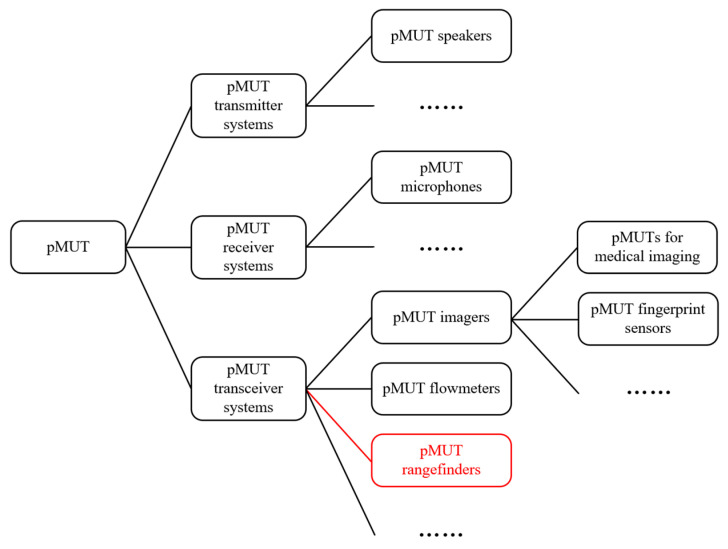
Different applications of pMUTs.

**Figure 2 micromachines-14-00374-f002:**
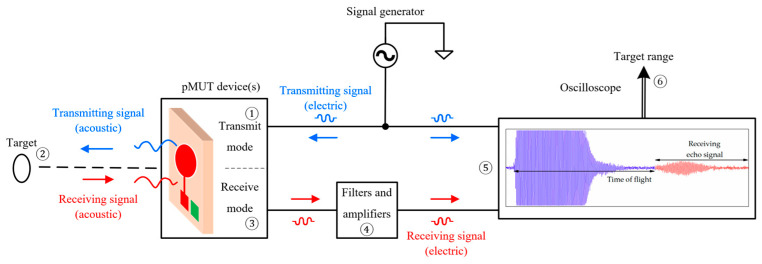
Block diagram of a pMUT rangefinder system. The signal shown in the oscilloscope is from [36] (Reproduced with permission from [36], Copyright 2018, MDPI AG).

**Figure 3 micromachines-14-00374-f003:**
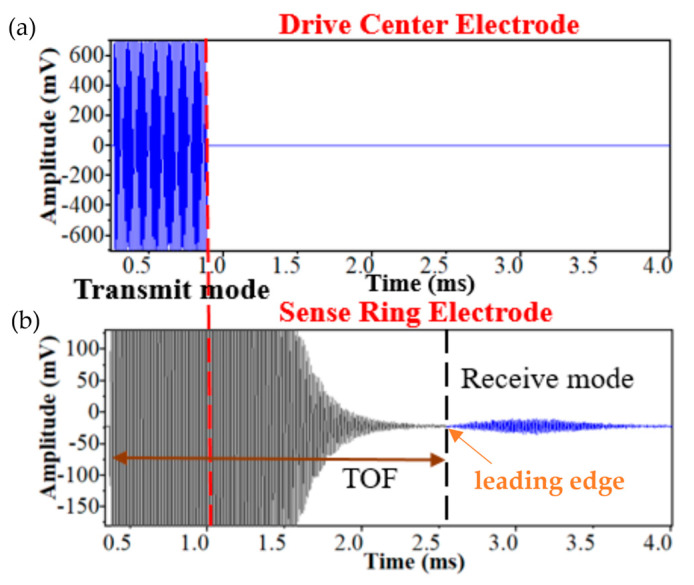
Illustration of the threshold TOF method. (**a**) The transmitting signals. (**b**) The receiving signal. Reproduced with permission from [39], Copyright 2019, IEEE.

**Figure 4 micromachines-14-00374-f004:**
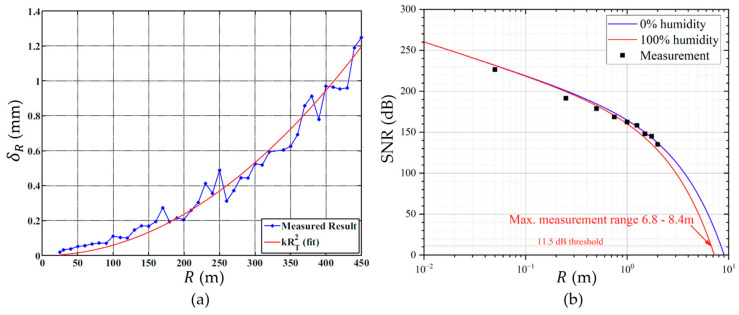
(**a**) $\delta_{R}$ versus R. Reproduced with permission from [33], Copyright 2010, IEEE. (**b**) SNR versus R. Reproduced with permission from [31], Copyright 2022, MDPI AG.

**Figure 5 micromachines-14-00374-f005:**
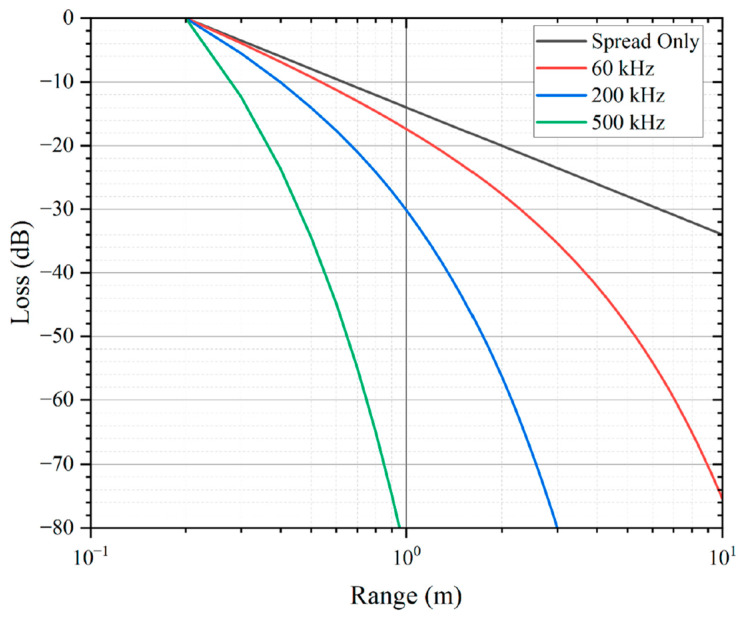
The transmission loss in a spread-only situation without absorption and in-air absorption situations with different frequencies at 20 °C and 50% air humidity. Reproduced with permission from [31], Copyright 2022, MDPI AG.

**Figure 6 micromachines-14-00374-f006:**
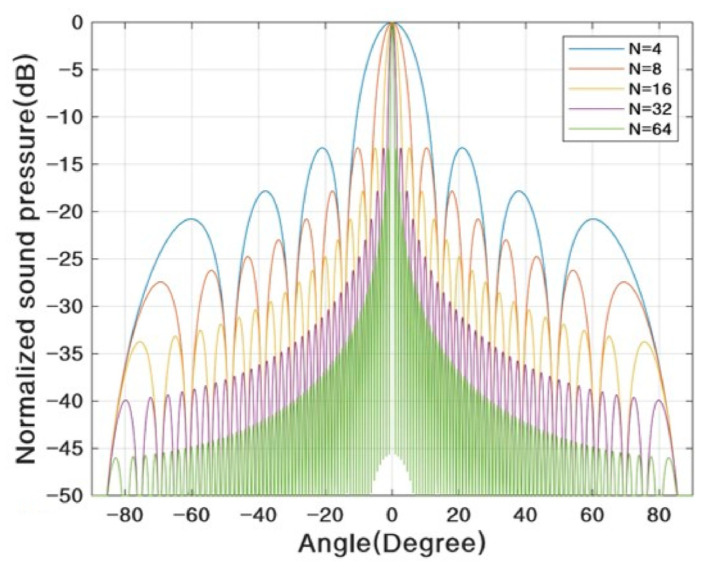
The influence of the element number on the directivity. Reproduced with permission from [58], Copyright 2022, MDPI AG.

**Figure 7 micromachines-14-00374-f007:**
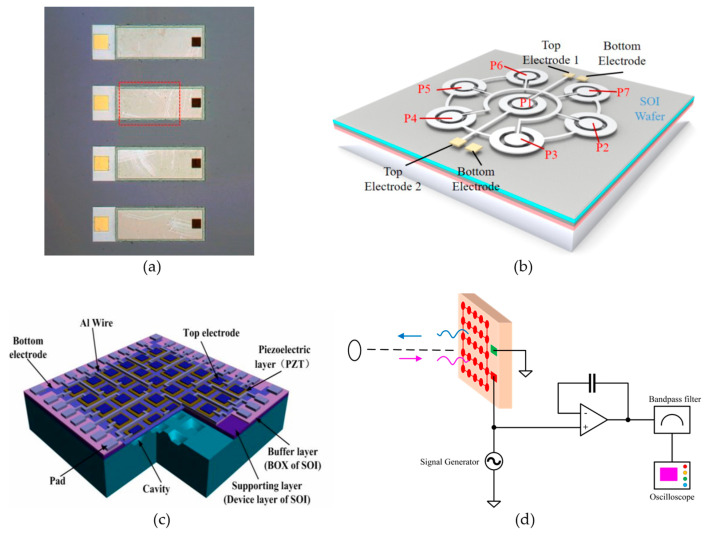
(**a**) A 1D pMUT array. Reproduced with permission from [61], Copyright 2016, IEEE. (**b**) A hexagonal pMUT array. Reproduced with permission from [62], Copyright 2021, IEEE. (**c**) A 2D square array. Reproduced with permission from [23], Copyright 2013, MDPI AG. (**d**) A pMUT rangefinder system using a pMUT array.

**Figure 8 micromachines-14-00374-f008:**
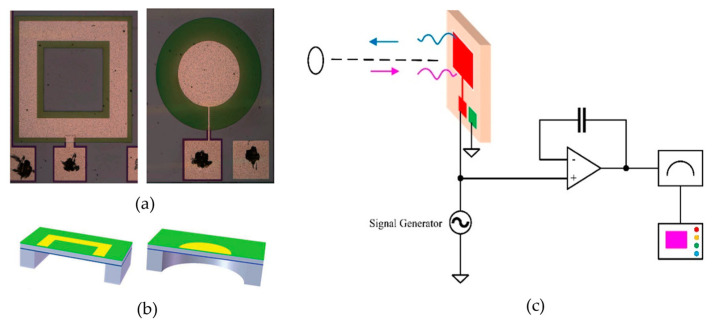
(**a**) Comparison between the structure of a square-membrane pMUT (**left**) and a circular-membrane pMUT (**right**). Reproduced with permission from [71], Copyright 2018, IEEE. (**b**) A 3D model of square- and circular-membrane pMUTs. Reproduced with permission from [71], Copyright 2018, IEEE. (**c**) A pMUT rangefinder system using a square-membrane pMUT.

**Figure 9 micromachines-14-00374-f009:**
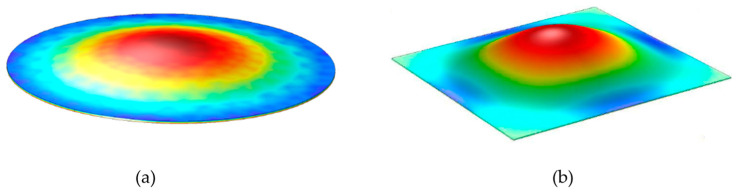
(**a**) The shape of deflection of a circular membrane. Reproduced with permission from [39], Copyright 2019, IEEE. (**b**) The shape of deflection of a square membrane. Reproduced with permission from [36], Copyright 2018, MDPI AG.

**Figure 10 micromachines-14-00374-f010:**
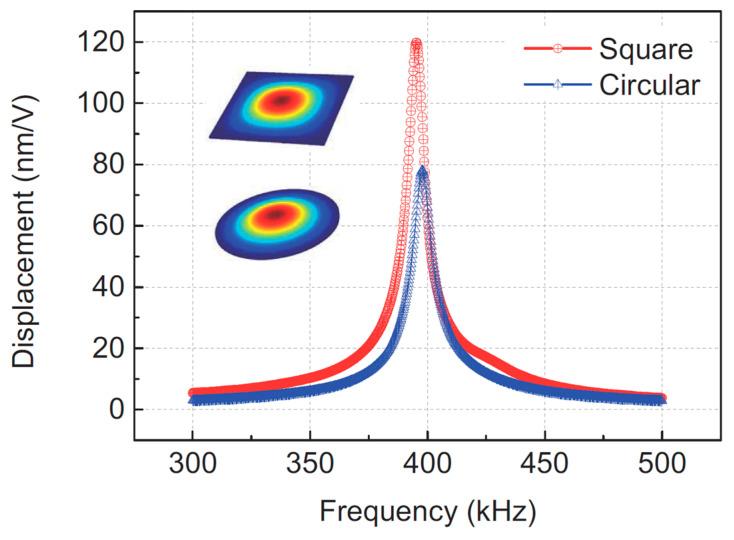
Tested vibration displacements of square- and circular-membrane pMUTs. Reproduced with permission from [71], Copyright 2018, IEEE.

**Figure 11 micromachines-14-00374-f011:**
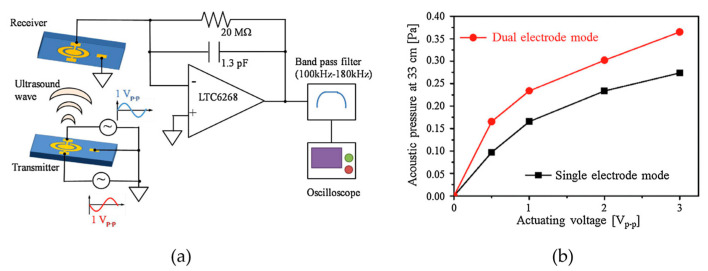
(**a**) A pMUT rangefinder using a dual electrode transmitter driven by anti-phase electric signals. (**b**) The relationship between the acoustic pressure and the driving voltage in dual electrode and single electrode modes. Reproduced with permission from [15], Copyright 2017, Elsevier B.V.

**Figure 13 micromachines-14-00374-f013:**
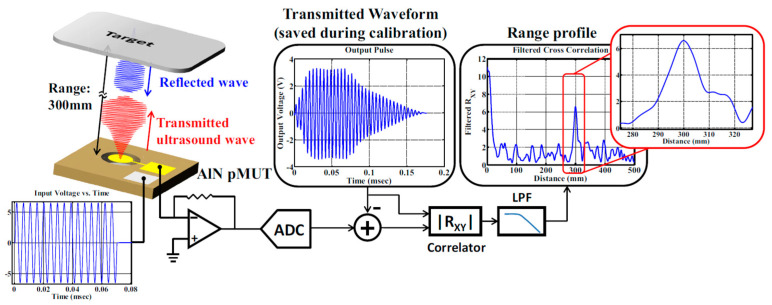
Block diagram of a pMUT rangefinder system using the cross correlation TOF method. Reproduced with permission from [33], Copyright 2010, IEEE.

**Figure 14 micromachines-14-00374-f014:**
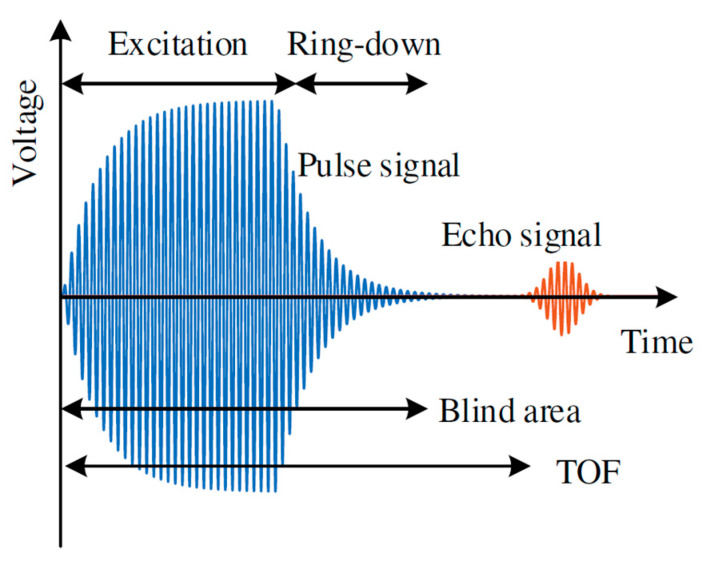
Illustration of the ring down of a pMUT rangefinder. Reproduced with permission from [107], Copyright 2021, MDPI AG.

**Figure 15 micromachines-14-00374-f015:**
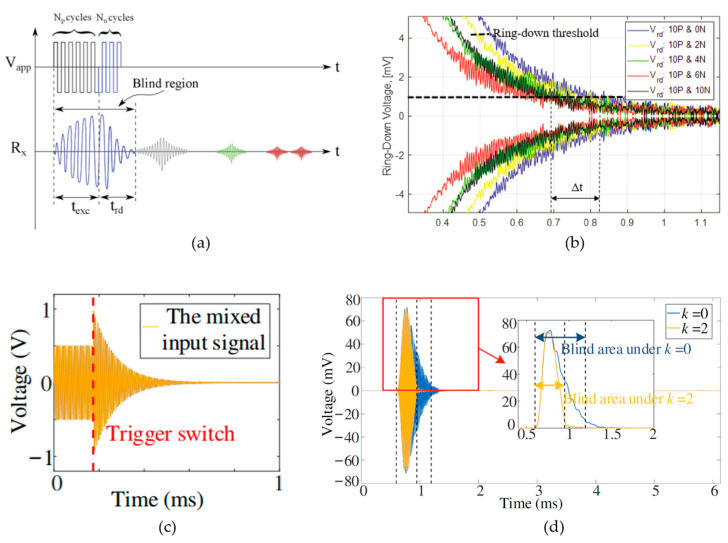
(**a**) Illustration of the 180° phase-shift ring-down suppression method [108]. Reproduced with permission from [108], Copyright 2021, IEEE. (**b**) The comparison experiment before and after suppression in [108]. Reproduced with permission from [108], Copyright 2021, IEEE. (**c**) The suppression signal of the ring-down suppression method is in [107]. Reproduced with permission from [107], Copyright 2021, MDPI AG. (**d**) The comparison experiment before and after suppression in [107]. Reproduced with permission from [107], Copyright 2021, MDPI AG.

**Figure 16 micromachines-14-00374-f016:**
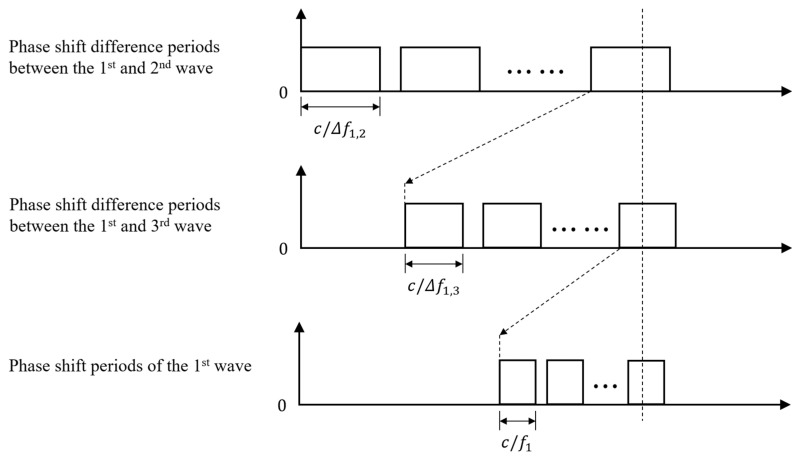
Illustration of the working mechanism of the MFCW method.

**Figure 17 micromachines-14-00374-f017:**
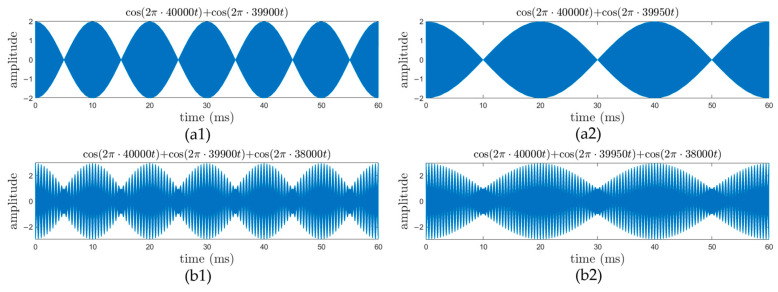
The illustration of TFCW waveforms (**a1**) Before period extension and (**a2**) After period extension, and MFCW waveforms (**b1**) Before period extension and (**b2**) After period extension.

**Figure 18 micromachines-14-00374-f018:**
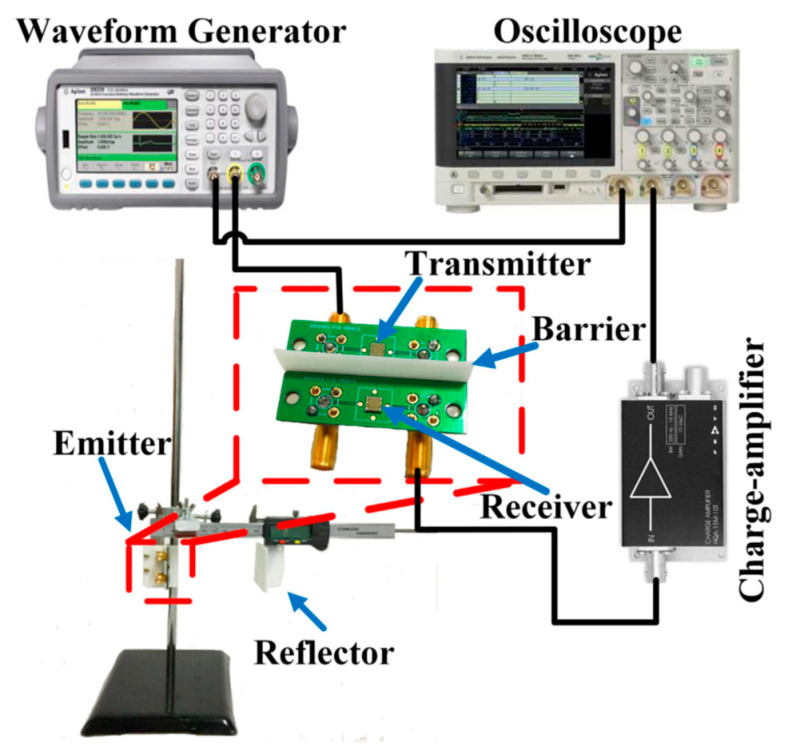
Block diagram of a MFCW pMUT rangefinder system. Reproduced with permission from [35], Copyright 2019, IEEE.

**Figure 19 micromachines-14-00374-f019:**
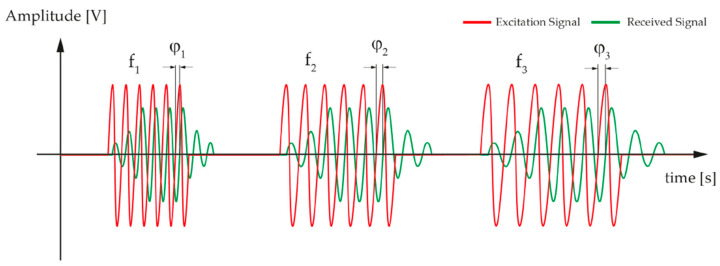
Illustration of the MFPW method. Reproduced with permission from [46], Copyright 2021, MDPI AG.

**Figure 20 micromachines-14-00374-f020:**
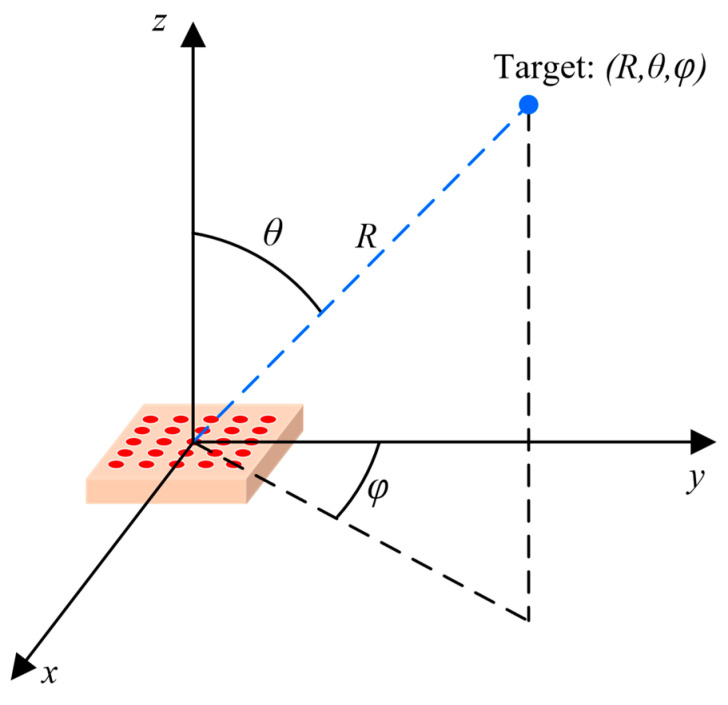
Illustration of a pMUT rangefinder target positioning system.

**Figure 21 micromachines-14-00374-f021:**
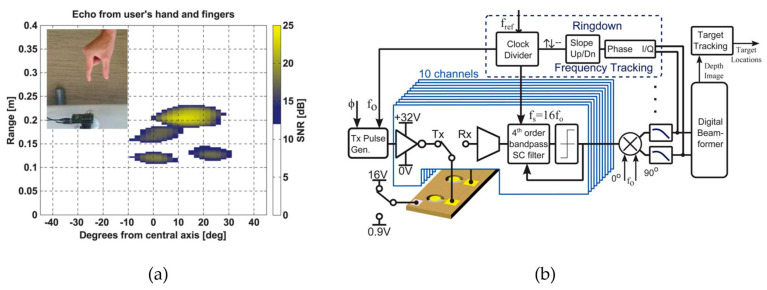
(**a**) A pMUT rangefinder object detection system for gesture recognition. (**b**) Block diagram of the pMUT rangefinder gesture recognition system. Reproduced with permission from [13], Copyright 2015, IEEE.

**Figure 22 micromachines-14-00374-f022:**
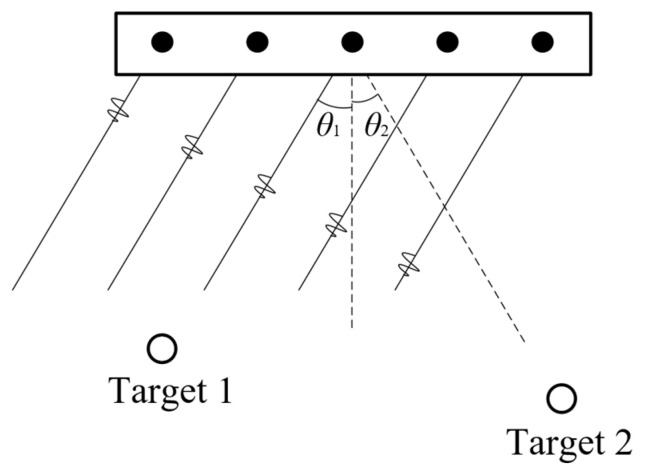
Illustration of steering during beamforming.

**Table 1 micromachines-14-00374-t001:** The signal form alternation during the signal propagation process.

Number	Signal Transmission Process	Corresponding Signals
①	Acoustic pressure generated by the pMUT transmit mode	$P_{tx}=G_{t}V_{d}$
②	Acoustic pressure received	$P_{rx}=P_{tx}\frac{R_{0}}{2R}G_{ac}e^{-2\alpha R}$
③	Electric signal generated by the pMUT receive mode	$V_{rx}=G_{r}P_{rx}$

**Table 2 micromachines-14-00374-t002:** Main factors that affect the performance of pMUT rangefinders.

Name	Symbol	Main Effect *
The driving voltage	$V_{d}$	$R_{max}$ ↑
The working frequency	f	Dual effects **
The resonant frequency	$f_{0}$	BW $↑\;\to R_{max}$ ↑ **
The transmitting sensitivity	$G_{t}$	$R_{max}$ ↑
The receiving sensitivity	$G_{r}$	$R_{max}$ ↑
The acoustic gain	$G_{ac}$	$R_{max}$ ↑
The quality factor	Q	Dual effects
The surface area of the membrane	A	$R_{max}$ ↑
The mean-square of the noise voltage	$\bar{n^{2}}$	$R_{max}$ ↓
Absorption coefficient	α	$R_{max}$ ↓

* “↑” means that the increase in this factor will enhance the parameter, and “↓” means that the increase in this factor will lower the parameter. ** f should be approximately equal to $f_{0}$ to achieve the maximum vibration, so their effects are not independent.

**Table 3 micromachines-14-00374-t003:** Summary of a single pMUT rangefinder or rangefinder arrays with different element numbers.

Element Number	$R_{max}$ (m) **	$\delta_{R}{}_{max}$ (mm) **	$SNR_{min}$	$V_{d}$ (V)	f (kHz)	$f_{0}$ (kHz)	Ref.
1 element	0.45	1.3	−	6.5	214	214	[69]
7 receiver elements *	0.75	3.5	28 dB (at 0.5 m)	15	−	190	[52]
7 elements	0.5	~0.67 ***	−	5	−	77.34	[62]
4 elements	2.4	−	12 dB	5	~48	−	[70]
14 elements	6.8	−	11.5 dB	5	66	66	[31]

* [52] used a line of seven elements in an array as the receiver units. ** All the ranges and the corresponding range errors that were measured by the distance between the transmitter and the receiver were then divided by two to be converted to the reflection mode results for performance comparison. *** The range error reported in [62] is seen as the 3σ error, which is divided by three to be converted to the RMS errors and then divided by two to be converted to the reflection mode results in this paper.

**Table 4 micromachines-14-00374-t004:** Summary of pMUT rangefinders with different device structures.

Structure Features	$R_{max}$ (m) *	$\delta_{R}{}_{max}$ (mm) *	$SNR_{min}$	$V_{d}$ (V)	f (kHz)	$f_{0}$ (kHz)	Ref.
Circular membrane	0.45	1.3	−	~5.9	214	214	[33]
Circular membrane	0.65	~0.28 **	~27.5 dB	13	−	215	[11]
Square membrane	0.5	0.63	−	1.8	97	97, 96	[72]
Square membrane	1.4	−	~0 dB	−	29	33	[36]
Dual electrode with anti-phase driving signals	1.1	−	12 dB	0.5	−	154	[15]

* All the ranges and the corresponding range errors that were measured by the distance between the transmitter and the receiver are divided by two to be converted to the reflection mode results for performance comparison. ** The 3σ error in [11] is divided by three to be converted to the RMS error and then divided by two to be converted to the reflection mode result.

**Table 6 micromachines-14-00374-t006:** Summary of pMUT rangefinders with different piezoelectric materials.

Piezoelectric Material	$R_{max}$ (m) *	$\delta_{R}{}_{max}$ (mm) *	$SNR_{min}$	$V_{d}$ (V)	f (kHz)	$f_{0}$ (kHz)	Ref.
AlN	0.45	1.3	−	6.5	214	214	[69]
PZT	0.0175	0.1225 **	−	1.5	~30	~30	[102]
PMnN-PZT	1.1	−	12 dB	0.5	−	154	[15]
Single-crystal PZT	2.4	−	12 dB	5	~48	−	[70]
AlScN	6.8	−	11.5 dB	5	66	66	[31]

* All the ranges and the corresponding range errors that are measured by the distance between the transmitter and the receiver are divided by two to be converted to the reflection mode results for performance comparison. ** The error was less than 0.7% in [102], so $\delta_{R}{}_{max}$ of [102] was estimated by 0.7%$R_{max}$.

**Table 7 micromachines-14-00374-t007:** Summary of rangefinders with continuous wave methods.

Device	Method	$R_{max}$/m *	$\delta_{R}{}_{max}$/mm *	Wave Frequencies	Ref.
Conventional ultrasonic transmitter and receiver	TFCW	0.1	1.5	39.85 and 40.6 kHz	[112]
Conventional ultrasonic transducers	MFCW	0.75	0.052	40, 39.9, 38.0 kHz	[111]
pMUTs	MFCW	0.3	~0.024 for R < 0.1 m, ~0.61 for 0.1–0.3 m **	497, 496.8, 487 kHz for R < 0.1 m 492, 491.8, 490 kHz for 0.1–0.3 m	[35]

* All the ranges and the corresponding range errors that were measured by the distance between the transmitter and the receiver were divided by two to be converted to the reflection mode results for performance comparison. ** The 3σ errors in [35] are divided by three to be converted to the RMS errors.

## Data Availability

Not applicable.

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
