# Peer review of "Review of Piezoelectric Micromachined Ultrasonic Transducers for Rangefinders"

_micromachines, 2023, doi:10.3390/mi14020374_

Round 1
Reviewer 1 Report
It is very meaningful to have a solid review paper on MEMS ultrasound technology. As the mushrooming of new emerging applications, it is definitely worthwhile to summarize the past research contribution and render the guidance for future development of PMUT device/system. However, there are some potential logic gaps in your explanation and the definition of some technical terms should match the standardized terminology to its content. For most of the table in this manuscript, the common baseline is not well defined before having a meaningful comparisons. The discussion on the optimization of device parameter trade-off, transducing configuration or algorithm can be more comprehensive and conclusive. In addition, there are many typos and wording issues, please revise your manuscript thoroughly.

Author Response
We appreciate Reviewer 1 for considering that it is very meaningful to have a solid review paper on MEMS ultrasound technology, and a past work summary and future look of pMUT device/system are worthwhile. In addition, your kind suggestions are very helpful for the improvement of this review paper, and the problematic parts that you have pointed out are carefully considered and corrected, including the discussions on device parameter trade-off, transducing configuration or algorithm, potential logic gaps, the definition of technical terms, etc. The main revisions of this manuscript are to rewrite and correct the principle of the pMUT rangefinder in Section 2, correct the misleading statements about some of the pMUT structures and add more information as well as corresponding reasoning about advanced pMUT devices that really contribute to the rangefinder improvement in Section 3, and supplement necessary information about ring-down suppression and continuous wave methods in Section 4.
Please see the attachment to check the point-by-point response, thanks again for your consideration!

Reviewer 2 Report
You have written a quite nice review. I appreciate your efforts. However, please try to improve the figures to make the paper more attractive to the community.
Also, you should consider citing "10.1109/ICEE50728.2020.9777041" somewhere in your text, since I personally consider this an important paper that tells readers about PMUT's acoustic field, something that has important relevance to ranging.
Author Response
We appreciate Reviewer 2 for your great support of our work. We have changed or adjusted most of the figures in this manuscript that are not reproduced by other publications, including Figures 1, 12, 17, 20, and 22. Most figures that are not changed are reproduced from other publications, and we think that reprinting them in the same form as the original publication can help readers of this paper more conveniently understand the meaning of relevant discussions.
Furthermore, we agree with you that the article "10.1109/ICEE50728.2020.9777041" is quite helpful for the discussion of this paper, and we have added a more comprehensive discussion during the analysis of directivity in Section 2 and have cited this article. Moreover, we have corrected the typos and English in this manuscript.
In addition, we have revised the content of the manuscript mainly in the following sections based on the suggestions from all the reviewers: We have corrected the analyses of the pMUT rangefinder principle and the relevant equations in Section 2. A more detailed discussion of the pMUT signal propagation processes is given in Section 2.1, and the analysis of pMUT rangefinder performance evaluation is provided in Section 2.2, where the reason to choose the maximum range and range error to evaluate the performance is discussed. Section 2.3 is added after Section 2.2 to discuss the main factors affecting pMUT rangefinder performance in detail, based on the analyses in the previous sections. Some of those factors can be improved by the advancement of pMUT devices or range-finding methods, which are described in Sections 3 and 4, respectively. Therefore, the overall logic of this paper is enhanced. In addition, the problematic analyses and expressions in Sections 3 and 4 that have been pointed out by the reviewers are revised, and more information is added as well. The advancements of pMUT device structures are combined and discussed in Section 3.2 in the revised manuscript. The analysis of the ring-down suppression is updated and collectively discussed in Section 4.1.
Thanks again for your consideration!
Reviewer 3 Report
Authors provided the review paper of pMUT rangefinders. Authors also showed future outlook for pMUT rangefinders. English grammar looks fine. There are some broken English expressions or grammar mistakes. However, authors showed good work for pMUT rangefinders. Therefore, the manuscript can be minor revision if authors answered the questions as below.
1. Please provide city and country information for conference papers.
2. Please use abbreviated journal names in reference section.
3. For some Figures, authors might need a permission for copyright.
4. Also-> In addition in Line 47. Please use formal English expression.
5. In Lne 98, Oscilloscope -> oscilloscope.
6. Please use Equation instead of Eq.
7. Please do not use thick fonts in Line 149-150.
8. Rangefinder Performance -> rangefinder performance.
9. Figure 12 labels seems to be small to be seen. Therefore, authors had better increase x-axis and y-axis label sizes.
10. Please provide ref. for the sentence (where ??=?0/? is the bandwidth) with ref. (https://www.sciencedirect.com/science/article/abs/pii/S0263224116306157)
11. In Figure 20, authors might need a permission from copyright if this is not a work from authors.
12. In Lines 660 and 661, please do not use thick and italic fonts about TOF.
Author Response
We thank Reviewer 3 for having precisely summarized the content and objective of this manuscript which are to provide the review paper of pMUT rangefinders and show future outlook for pMUT rangefinders. Moreover, your kind suggestions are really helpful for the improvement of this manuscript, and we have revised our manuscript according to your suggestions.
We have revised all your suggestions, and please see the attachment to check the point-by-point response. Thanks again for your consideration!

Round 2
Reviewer 1 Report
None